# Characterization of the Key Bibenzyl Synthase in *Dendrobium sinense*

**DOI:** 10.3390/ijms23126780

**Published:** 2022-06-17

**Authors:** Yan Chen, Yu Wang, Chongjun Liang, Liyan Liu, Xiqiang Song, Ying Zhao, Jia Wang, Jun Niu

**Affiliations:** Key Laboratory of Genetics and Germplasm Innovation of Tropical Special Forest Trees and Ornamental Plants-Ministry of Education, College of Forestry, Hainan University, Haikou 570228, China; chenyan1520898@sina.com (Y.C.); wangy1998073@163.com (Y.W.); chongjun_liang@163.com (C.L.); lly13101@163.com (L.L.); songstrong@hainanu.edu.cn (X.S.); zhaoying3732@163.com (Y.Z.)

**Keywords:** *Dendrobium sinense*, bibenzyl biosynthesis, bibenzyl synthase, prokaryotic expression

## Abstract

*Dendrobium sinense*, an endemic medicinal herb in Hainan Island, is rich in bibenzyls. However, the key rate-limited enzyme involved in bibenzyl biosynthesis has yet to be identified in *D. sinense*. In this study, to explore whether there is a significant difference between the *D. sinense* tissues, the total contents of bibenzyls were determined in roots, pseudobulbs, and leaves. The results indicated that roots had higher bibenzyl content than pseudobulbs and leaves. Subsequently, transcriptomic sequencings were conducted to excavate the genes encoding type III polyketide synthase (PKS). A total of six *D. sinense PKS* (*DsPKS*) genes were identified according to gene function annotation. Phylogenetic analysis classified the type III *DsPKS* genes into three groups. Importantly, the c93636.graph_c0 was clustered into bibenzyl synthase (BBS) group, named as *D. sinense* BBS (DsBBS). The expression analysis by FPKM and RT-qPCR indicated that *DsBBS* showed the highest expression levels in roots, displaying a positive correlation with bibenzyl contents in different tissues. Thus, the recombinant DsBBS-HisTag protein was constructed and expressed to study its catalytic activity. The molecular weight of the recombinant protein was verified to be approximately 45 kDa. Enzyme activity analysis indicated that the recombinant DsBBS-HisTag protein could use 4-coumaryol-CoA and malonyl-CoA as substrates for resveratrol production in vitro. The Vmax of the recombinant protein for the resveratrol production was 0.88 ± 0.07 pmol s^−1^ mg^−1^. These results improve our understanding with respect to the process of bibenzyl biosynthesis in *D. sinense*.

## 1. Introduction

*Dendrobium sinense*, belonging to the orchid family, is a kind of endemic medicinal herb in Hainan Province [1]. This species is distributed over the tropical montane rainforest of central and western Hainan Island, such as Baoting, Qiongzhong, Ledong, and Baisha County. Historically, numerous properties of pharmacology of the *Dendrobium* genus were highly appreciated in China and Southeast Asian countries. Phytochemical analysis has also showed varied bioactive constituents in *Dendrobium* plants [2]. The studies of *Dendrobium* species tend to focus on exploration and pharmacology of bibenzyls. For example, 4,5-dihydroxy-3,3,4-trimethoxybibenzyl isolated from *D. lindleyi* has an inhibitory effect on lung cancer growth and metastasis [3]. The bibenzyl components from *D. falconeri* could promote the expression of integrin, which inhibits epithelial-mesenchymal transition to block the migration and proliferation of lung cancer cells [4]. In previous investigations, it was found that bibenzyl was one of the essential phenolic ingredients in *D. sinense* [5]. The pharmacological effect of bibenzyl indicated that four bibenzyls possessed good inhibitory activity against various human cancer cell lines [6]. Previous studies in *D. sinense* mainly focused on the extraction, structures, and pharmacological activities of bibenzyls, but the key regulatory genes of biosynthetic pathways for bibenzyl have yet to be identified.

Bibenzyls are aromatic compounds with common parent nucleus, consisting of two phenyls linked with ethane (C6-C2-C6) [7]. Depending on the number of repeats of parent nucleus, bibenzyls can be divided into two classes, namely simple and double bibenzyl [8]. Although this structure of parent nucleus is simple, the different substitutions (such as methyl, methoxyl, hydroxyl, glycosyl, and chlorine) on the bridge chain and aromatic rings result in a variety of bibenzyl compounds [9]. Bibenzyls are a class of secondary metabolites in plants, but these natural compounds are mainly concentrated in bryophytes and *Dendrobium* plants [7,8].

Bibenzyls, belonging to the polyketide family, are derived from the pathway of phenylpropanoid biosynthesis. This pathway is responsible for the biosynthesis of many secondary metabolites [10]. The different directions of phenylalanine metabolism have great influence on the accumulation of secondary metabolites and plant cell development [11]. Phenylalanine is first deaminized to form cinnamic acid, coumaric acid, and other acids with a phenylpropane (C6-C3) unit. Subsequently, the CoA esters of these acids are catalyzed to product 4-coumaroyl-CoA and dihydro-4-coumaroyl-CoA, which are initial substrates for polyketide biosynthesis [12]. In addition to the initial substrates, malonyl-CoA is also an important compound for polyketide chain elongation.

Type III polyketide synthases (PKSs) catalyze cyclization and aromatization of intermediate to form polyketides [13]. The combinational difference of enzyme bound substrates may cause the functional difference of PKSs [14]. For example, chalcone synthase (CHS) as a member of the PKS family catalyzes the Claisen cyclization to produce chalcones and dihydrochalcones, which are essential precursors for flavonoid biosynthesis [15]. Benzylacetone synthase (BAS) catalyzes the production of benzylacetone derivatives [16]. In orchid species, it is worth noting that bibenzyl synthase (BBS) was first isolated and purified from *Bletilla striata*, and its aldol-type cyclized activity was verified using in vitro enzyme assay [17]. Additionally, pBibSy811 and pBibSy212 were cloned from *Phalaenopsis* plants, and the encoded proteins could catalyze aldol cyclization [18]. To date, about 2045 type III *PKS* genes have been reported in the NCBI database, but it is still not known which key *BBS* genes are responsible for bibenzyl biosynthesis in *D. sinense*.

In this study, the total content of bibenzyls in roots, pseudobulbs, and leaves of *D. sinense* was first measured. Subsequently, the transcriptomic sequencing of the samples from roots, pseudobulbs, and leaves of *D. sinense* was performed by Illumina platform. All of *D. sinense PKS* (*DsPKS*) genes were identified according to the gene annotation results. Phylogenetic analysis classified the *DsPKS* genes into subclusters. The expression levels of the crucial *PKS* genes were verified by real-time quantitative polymerase chain reaction (RT-qPCR). Combined with the results of gene expressions and bibenzyl contents, the *D. sinense BBS* (*DsBBS*) genes were characterized by correlation analysis. After gene clone and protein expression, the catalytic activity of *DsBBS* genes was identified by in vitro enzymatic assay. This study revealed the key *DsBBS* gene involved in bibenzyl biosynthesis of *D. sinense*, providing the basis for further understanding the biosynthetic mechanism of bibenzyls.

## 2. Results

### 2.1. Total Contents of Bibenzyls in Different Tissues of D. sinense

Healthy and similar tissue-cultured plantlets were selected for follow-up analysis (Figure 1a). To explore whether there was a statistically significant difference between tissues, the total contents of bibenzyls were detected in the roots, pseudobulbs, and leaves of *D. sinense*, respectively. Based on the results of standard curve, it was calculated that the total contents of bibenzyls accounted for 1.31%, 0.62%, and 0.72% of the dry weight of roots, pseudobulbs, and leaves, respectively (Figure 1b). Statistical analysis showed that the bibenzyl content in roots was significantly greater (*p* < 01) than that in pseudobulbs and leaves, while no significant difference between pseudobulbs and leaves was observed (Figure 1b).

### 2.2. Transcriptome Sequencing and Analysis

Due to the lack of genomic information, transcriptomic libraries constructed from roots, pseudobulbs, and leaves were sequenced by Illumina platform to obtain gene data of *D. sinense*. After sequencing quality control, a total of 58.10 Gb clean data were obtained (Appendix A). The transcriptomic data of *D. sinense* have been committed to the NCBI database, such as SRR15112264, SRR15112265, and SRR15112266 for leaves, SRR15112267, SRR15112268, and SRR15112269 for pseudobulbs, and SRR15112270, SRR15112271, and SRR15112272 for roots.

The high-quality clean reads were assembled by the Trinity software. The resulting 61,180 unigenes (mean: 1000.25 bp) were obtained. By comparison with homology proteins in databases, 13,248 (21.65%), 9287 (15.18%), 17,503 (28.61%), 17,047 (27.86%), and 28,683 (46.88%) gene were functionally annotated in GO, KEGG, Pfam, SwissProt, and Nr databases, respectively. These gene resources of *D. sinense* will serve as a base for future studies.

### 2.3. High-Throughput Analysis of Gene Expression

Digital gene expression was calculated by FPKM. According to the results of digital gene expressions, the assessment of correlation between samples was performed using the Pearson correlation coefficient algorithm. Compared with intergroup samples, intragroup samples showed a higher correlation (Figure 2a). Indeed, PCA analysis displayed that the discovery cohort clustered separately among roots, pseudobulbs, and leaves (Figure 2b), suggesting notable variation and good repeatability in the intergroup and intragroup of our samples, respectively.

The Benjamini–Hochberg algorithm with an adjusted *p*-value < 0.01 and |log_2_ (fold-change)| > 2 was used to find the DEGs between different tissues. There were 2331 DEGs in roots vs. pseudobulbs, including 1384 upregulated genes and 927 downregulated genes (Figure 2c). Moreover, 3527 DEGs (2003 upregulation and 1524 downregulation) and 3016 DEGs (1694 upregulation and 1372 downregulation) were identified in roots vs. leaves and pseudobulbs vs. leaves, respectively (Figure 2c).

### 2.4. Identification of the Type III PKS Genes in D. sinense

Based on the results of functional domain and gene annotation, a total of six type III *PKS* genes were obtained. By ORF analysis, *DsCHS2* (c82700.graph_c0), *DsCHS3* (c84946.graph_c0), *DsPKS* (c89783.graph_c0), and *DsBBS* (c93636.graph_c0) were characterized with a complete ORF, the lengths of which were 1188 (395 aa), 1173 (390 aa), 1104 (337 aa), and 1173 nt (390 aa), respectively (Table 1).

To explore the genetic cluster of type III PKSs, these proteins were obtained from *A. thaliana*, *D. sinense*, and *D. catenatum*. The results of the evolutionary tree showed that all *PKS* genes were divided into three categories, namely CHS group, BBS group, and PKS group (Figure 3). The type III *PKS* genes in *D. sinense* were named according to the results of genetic cluster. The genes of c84946.graph_c0 (*DsCHS1*), c82700.graph_c0 (*DsCHS2*), c99839.graph_c0 (*DsCHS3*), and c82697.graph_c0 (*DsCHS4*) belonged to the CHS group (Figure 3). The c89783.graph_c0 (*DsPKS*) was in the PKS group (Figure 3). Importantly, the protein encoded by c93636.graph_c0 (*DsBBS*) was clustered into BBS group with other BBS proteins from *D. catenatum* (Figure 3).

### 2.5. Expression Analysis

To further explore the key type III *PKS* gene involved in bibenzyl biosynthesis, the expression profiles of type III *PKS* genes were concretely calculated by FPKM. A heat map was demonstrated in colors that reflect expression levels; the lowest expression level was indicated in green, and the highest expression level was indicated in red (Figure 4a). The *DsCHS1*, *DsCHS4*, and *DsPKS* genes showed no or low expression levels in roots, pseudobulbs, and leaves of *D. sinense* (Figure 4a). Additionally, *DsCHS2* and *DsCHS3* genes exhibited significant higher expressions in pseudobulbs (Figure 4a). Interestingly, the highest *DsBBS* expression levels were found in *D. sinense* roots (Figure 4a). More surprising was the apparent correlation of *DsBBS* expression with bibenzyl content. To accurately prove its expression level, the expression levels of *DsBBS* were analyzed by RT-qPCR. Indeed, the expressions of *DsBBS* gene in roots were higher than in pseudobulbs and leaves (Figure 4b). These expression results suggested that the DsBBS (c93636.graph_c0) may be the key rate-limiting factor in bibenzyl biosynthesis.

### 2.6. High Expression of Recombinant DsBBS Protein

To obtain high-quality recombinant protein, it is necessary to screen the conditions of protein expression and solubility. To achieve high protein expression, the recombinant DsBBS-HisTag was induced in 0.1, 0.3, 0.5, 0.8, and 1.0 mmol of isopropyl-*β*-d-thiogalactopyranoside (IPTG). It was clear that the recombinant DsBBS-HisTag protein was poorly expressed under the uninduced condition (control, 0 mmol of IPTG), while the recombinant DsBBS-HisTag protein could be expressed with high yields in all solubility (Figure 5a). Notably, the protein content induced by 1.0 mmol/L IPTG was the highest. Under different temperature and time of induction, the recombinant DsBBS-HisTag protein was mainly concentrated in precipitate at 37 °C, whereas the recombinant protein was rich in supernatant at 15 °C (Figure 5b). Thus, this condition (1.0 mmol of IPTG, 15 °C, 24 h) can be used for purification of soluble proteins.

### 2.7. Enzyme Activity Analysis

Using affinity purification, the recombinant DsBBS-HisTag protein was purified and analyzed by SDS-PAGE. The band of recombinant DsBBS-HisTag protein was in line with theoretical molecular weight of 44.33 kDa (Figure 6a). The 4-coumaroyl-CoA and dihydro-4-coumaroyl-CoA are commonly used as a shared substrate for the biosynthesis of bibenzyls and flavonoids (Figure 6b). To explore the catalytic functions of DsBBS protein, the enzyme activity analysis was performed. After incubation, it was found that one product shared the same HPLC retention time as resveratrol (Figure 6c). This result indicated the high substrate specificity of DsBBS protein for 4-coumaryol-CoA. Interestingly, the chromatograms of in vitro enzyme assays also showed that the recombinant DsBBS protein specifically catalyzes the cyclization and aromatization of 4-coumaryol-CoA and malonyl-CoA to generate bibenzyls (Figure 6b, orange line). The Vmax of the recombinant DsBBS-HisTag protein for the resveratrol production was 0.88 ± 0.07 pmol s^−1^ mg^−1^.

## 3. Discussion

*D. sinense* is a tropical epiphytic plant endemic to Hainan Island with great ornamental and medicinal value, similar to other *Dendrobium* plants in China [19]. Component and pharmacology analysis indicated that plants from this genus are rich in aromatic compounds, containing a large number of bibenzyl compounds [5]. Compared with other *Dendrobium* plants, the bibenzyl content in *D. sinense* was higher, and three bibenzyl compounds had better nematocidal activity [20]. Four bibenzyl compounds from *D. sinense* exhibited strong inhibitory activity against various human cancer cell lines [21]. Previous component analysis of *D. sinense* has shown that the whole plant is rich in bibenzyls, but there is a question as to whether there is a significant difference in bibenzyl content between the different tissues. It is apparent from our results that *D. sinense* roots are particularly rich in bibenzyls compared with pseudobulbs and leaves (Figure 1b). This difference implies that the strength of bibenzyl biosynthesis varies between different tissues. A possible explanation for this might be that the key limiting enzyme involved in bibenzyl biosynthesis shows differential expression in different tissues. This hypothesis was confirmed by FPKM and RT-qPCR (Figure 4).

With the advent of high-throughput sequencing technologies, it has become increasingly feasible to explore a single gene or gene family [1,13,22]. To identify the key type III *PKS* genes involved in bibenzyl biosynthesis, the roots, pseudobulbs, and leaves of *D. sinense* were individually sequenced by Illumina platform. After gene assembling and functional annotation, a total of six type III *PKS* genes were excavated and identified. The type III PKS catalyzes iterative decarboxylative condensations of malonyl-CoA with a CoA-linked starter molecule, generating abundant polyketides such as chalcones, stilbenes, acridones, and bibenzyls [23]. By phylogenetic analysis, all type III *PKS* genes were clustered into three groups (Figure 3). The CHS group contains four *D. sinense* genes of *DsCHS1*, *DsCHS2*, *DsCHS3*, and *DsCHS4*, while the PKS and BBS group each contain one gene (Figure 3). A highly similar gene structure is exhibited in the same phylogenetic cluster, suggesting that the homology genes may have the same function [24]. Thus, the *DsBBS* (c93636.graph_c0) gene has received considerable attention in this study.

To further evaluate the functions of the six type III *PKS* genes in *D. sinense*, their expression levels were analyzed by FPKM and RT-qPCR. Different type III *PKS* genes showed various expression profiles in different tissues (Figure 4a), suggesting that they may have different functions in *D. sinense*. We observed that the FPKM of *DsBBS* (c93636.graph_c0) gene was significantly higher in roots than in pseudobulbs and leaves, which was confirmed by RT-qPCR (Figure 4). Overall, combined with the results of phylogenetic analysis and expression profiles, it is tempting to speculate that DsBBS might serve as an important rate-limiting enzyme in bibenzyl biosynthesis.

To explore the enzymatic activity of DsBBS, the recombinant DsBBS-HisTag protein was constructed and expressed. After electrophoresis, the band of recombinant DsBBS-HisTag was at about 45 kDa, which was similar to the molecular weight of BBS (46 kDa) in *Bletilla striata* [17]. Although multiple substrates can be used for polyketide biosynthesis, such as phenylpropionyl-CoA, cinnamoyl-CoA, *m*-coumaroyl-CoA, dihydro-*m*-coumaroyl-CoA, *p*-coumaroyl-CoA, and dihydro-*p*-coumaroyl-CoA, *p*-coumaroyl-CoA (4-coumaroyl-CoA) is the most common substrate as the starter molecule of polyketides in plants [25]. Through the phenylpropane metabolic pathway, 4-coumaroyl-CoA and dihydro-4-coumaroyl-CoA are used as common substrates for flavonoid and bibenzyl biosynthesis [23]. The HPLC results indicated that the catalytic activity of DsBBS was specific, using 4-coumaroyl-CoA as the substrate for resveratrol production. The Vmax of DsBBS-HisTag protein was 0.88 pmol s^−1^ mg^−1^ for resveratrol generation, which was slightly higher than 0.68 pmol s^−1^ mg^−1^ of PpCHS in *Physcomitrella patens* [26] and 0.46 pmol s^−1^ mg^−1^ of PlCHS in *Pueraria lobata* [27].

## 4. Materials and Methods

### 4.1. Plant Materials

The tissue culture seedlings of *D. sinense* were cultured at Hainan University, Haikou, China as stated in the previous report [28]. Healthy tissue-cultured plantlets were selected with uniform growth. The selection criteria were 5–6 cm in height, 4–5 roots, and 5–6 leaves. The fresh roots, pseudobulbs, and leaves of *D. sinense* were collected from five tissue culture seedlings. The *D. sinense* samples were put into prechilled tubes and stored at −80 °C.

### 4.2. Identification of Bibenzyl Content

Determination of bibenzyl content was performed basically as described before with slight modifications [29]. Briefly, the roots, pseudobulbs, and leaves of *D. sinense* were dried at 85 °C to achieve a constant weight. A total of 100 mg sample powder was refluxed with 70% ethanol (2 mL) at 90 °C for 2 h. The sample was then centrifuged at 12,000 rpm for 10 min. The supernatant of extract was transferred to new centrifugal tubes. After concentration by rotary evaporation, methanol was added to 10 mL volumetric flask. The gigantol was used as standard to map the standard curve. The absorption values of bibenzyl compounds were determined by ultraviolet spectrophotometer (METASH, Shanghai, China) at 280 nm. Based on the standard curve, the concentration of bibenzyl compounds was calculated by standard curve. The experiment was performed three times.

### 4.3. Transcriptome Preparation and Sequencing

The extraction of *D. sinense* RNA was conducted following manufacturer’s instructions (Qiagen, Frankfurt, Germany). The subsequent preparation of transcriptome library for high-throughput sequencing was performed as in [30]. These constructed libraries were sequenced using Illumina platform as in [31]. The generated data were processed by BMKCloud online platform v2.0 (www.biocloud.net, accessed on 14 February 2022) to obtain clean reads. Trinity software was used for gene assembly with all parameters set to default [32].

### 4.4. Functional Annotation and Expression Analysis

The gene functions were annotated based on the public databases, as described in our previous study [1]. Bowtie software was used to compare the clean reads with unigene database. Using the algorithm of fragments per kilobase per million (FPKM), gene expressions were estimated by RSEM software [33]. Pearson correlation coefficient and principal component analysis (PCA) were computed by R Project. Differentially expressed genes (DEGs) between two groups were identified using DESeq. The *P* value of significant difference was adjusted using the previous approach for controlling the false discovery rate [34]. DEGs were identified by a *p*-adjust < 0.01 and |log_2_ (expression-fold)| > 2.

### 4.5. Identification of Type III PKS Genes and Phylogeny Tree

The homologous protein sequences of *Arabidopsis thaliana* PKS (AtPKS) and *D. catenatum* PKS (DcPKS) were acquired from TAIR and NCBI Genome Data Viewer ASM160598v2 (ncbi.nlm.nih.gov/genome/gdv/, accessed on 27 February 2022), respectively. To comprehensively define the type III *PKS* genes in *D. sinense*, these protein sequences were used to screen type III *PKS* genes (E value < e^−10^) in addition to the results of functional annotation. The open reading frame of DsPKSs was identified by BioEdit software. All protein sequences of the type III PKS were aligned using ClustalW with default options. The neighbor-joining tree was generated with MEGA7, using the previous parameters [35].

### 4.6. RT-qPCR Analysis

Based on the transcriptome sequences of *D. sinense*, RT-qPCR primers of the key *DsPKS* genes were designed by PrimerQuest™ Tool (Appendix A). According to our previous investigation, actin-depolymerizing factor 11 (ADF11) and acyl-CoA binding protein 2 (ACBP2) were identified to be the two best stable references in different *D. sinense* tissues (unpublished data). Thus, the two genes were used as internal references in this study. The RT-qPCR and melting curve procedure were performed by Lightcycler 96 (Roche, Penzberg, Germany). The 20 μL RT-qPCR reaction system was prepared using the MonAmp™ ChemoHS qPCR Mix (SYBR Green I) Kit (Monad, Guangzhou, China). The amplification procedures started with an initial step of 95 °C for 30 s, followed by 40 cycles of denaturation at 95 °C for 5 s, annealing at 55 °C for 10 s, and extension 72 °C for 30 s. A melting curve procedure was conducted to validate the unitary PCR product. Experiments were repeated a minimum of three times. The expression levels were calculated by 2^−ΔΔCt^.

### 4.7. Gene Cloning and Recombinant Vector Construction

Based on the transcriptome sequences of *D. sinense*, the open reading frames (ORF) of *DsPKS* genes were amplified from cDNA using the full-length primers (Appendix A). The amplified products were linked to the pCloneEZ-TA (ThBio, Beijing, China) and then sequenced by Sangon Biotech (Guangzhou, China). To correctly construct DsPKS-HisTag fusion expression vector, the downstream primer was redesigned to delete the termination codon of *DsPKS* genes. In addition, the terminal homologous sequence (15 bp) of pET28a vector was added into the 5′ end of upstream and downstream primers (Appendix A). The pET28a vector was digested by *Xba*I and *Sac*I (NEB, Beijing, China). The DsBBS gene was inserted into the linearized pET28a vector using the ClonExpress^®^ II One Step Cloning Kit (Vazyme, Nanjing, China). The transformation vector of DsBBS-HisTag was introduced into *Escherichia coli* strain BL21(DE3) for protein expression.

### 4.8. Protein Purification and Enzymatic Assay

The *E. coli* bacteria were cultured in 1 L conical flask at 37 °C with 175 rpm shaking. When the optical density value reached 0.5~0.6, the recombinant DsBBS-HisTag protein was induced using different solubility of IPTG at 37 °C for 6 h, 23 °C for 16 h, and 15 °C for 24 h [36]. After centrifugation at 4 °C for 20 min, the precipitum was resuspended in 150 mL 10 mmol/L imidazole buffer, and then sonicated on an ice-water mixture for 20 min by JY92-IIN sonicator (Xinzhi, Ningbo, China). Subsequently, centrifugation of the resuspension was performed at 4 °C for 20 min. Using Ni-IDA resin (Novagen, Madison, WI, USA), the supernatant was purified. The purified DsBBS-HisTag protein was detected with SDS-PAGE and visualized by staining with Coomassie brilliant blue staining.

To verify the activity of recombinant protein, 50 μL volume of enzymatic reaction, including 10 μg DsBBS-HisTag fusion protein, 50 mM Hepes buffer pH 7.0 (Biosharp, Beijing, China), 0.5 mM malonyl-CoA lithium salt (Yuanye, Shanghai, China), and 0.5 mM 4-coumaroyl CoA (Yuanye, Shanghai, China) were incubated at 37 °C for 30 min. Finally, 50 μL volume of methanol was added to terminate the reaction. After centrifugation at 12,000 rpm for 10 min, the 20 μL supernatant was analyzed by high-performance liquid chromatography (HPLC).

### 4.9. HPLC Analysis

To detect the target, the samples were analyzed by an LC-100 PUMP (Wufeng, Shanghai, China) equipped with HC-C18 column (18 μm, 4.6 × 250 mm, Agilent, Santa Clara, CA, USA). The chromatographic conditions were as follows: column temperature 30 °C; injection sample volume 20 μL, solution A 0.1% phosphate water; solvent B 100% acetonitrile; flow rate 1 mL/min. The gradients for the solvents were 0–6 min, 70% solution A, 30% solvent B; 6–40 min, 55% solution A, 45% solvent B. The standard substances of resveratrol, dihydroresveratrol, phloretin, and chalconaringenin were purchased from Yuanye Bio-Technology Company, Shanghai, China. Comparison of retention time with the standard was used for determination.

## 5. Conclusions

By the analysis of the bibenzyl content in different tissues, the *D. sinense* roots had higher bibenzyl content than pseudobulbs and leaves. Based on transcriptome data of roots, pseudobulbs, and leaves, a total of six type III *PKS* genes were annotated, of which one (c93636.graph_c0 named as *DsBBS*) was clustered into BBS group by phylogenetic analysis. Additionally, expression analysis suggested that the expression patterns of *DsBBS* gene exhibited a positive correlation with bibenzyl contents in different tissues. Prokaryotic expression vector containing *DsBBS* gene was constructed. The band of recombinant DsBBS-HisTag protein was in line with theoretical molecular weight of 44.33 kDa. Enzyme activity analysis indicated that the recombinant protein could catalyze the Claisen cyclization to produce bibenzyls. The Vmax of the recombinant protein for the resveratrol production was 0.88 ± 0.07 pmol s^−1^ mg^−1^. Although the biosynthesis pathway of bibenzyls in *D. sinense* remains largely unknown, the key *DsBBS* gene encoding the rate-limiting enzyme involved in bibenzyl biosynthesis has been identified. These findings would be conducive to understanding the underlying mechanisms of bibenzyl biosynthesis in the orchid family.

## Figures and Tables

**Figure 1 ijms-23-06780-f001:**
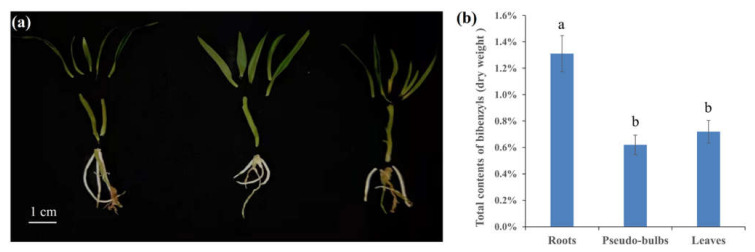
The images and bibenzyl contents in roots, pseudobulbs, and leaves of *D. sinense*. (**a**) The roots, pseudobulbs, and leaves of *D. sinense*. Scale bar, 1 cm. (**b**) The total content of bibenzyls in different *D. sinense* tissues. Data are reported as mean ± standard deviation. Different letters mean significant difference (*p* < 0.01).

**Figure 2 ijms-23-06780-f002:**
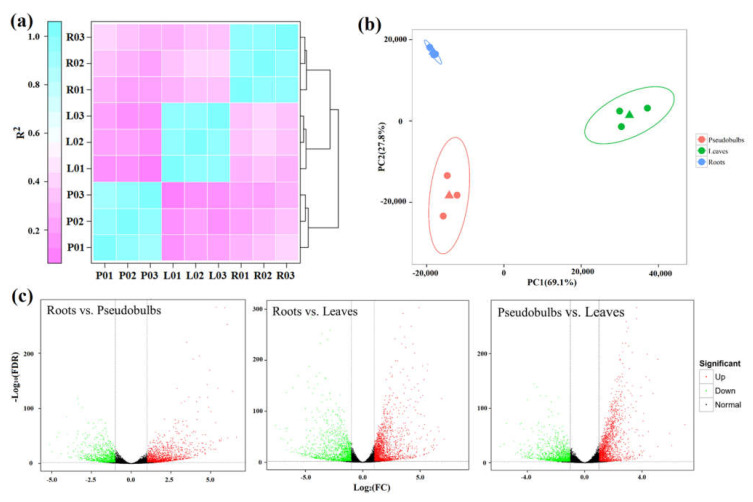
Analysis of differential gene expression in different *D. sinense* tissues. (**a**) Pearson correlation coefficient of transcriptome samples. The greater the value, the greater the correlation. R, roots; P, pseudobulbs; L, leaves. (**b**) Principal component analysis plot. The circles represent different sequencing samples, and triangles represent the average values. (**c**) The analysis of differential gene expression in different groups.

**Figure 3 ijms-23-06780-f003:**
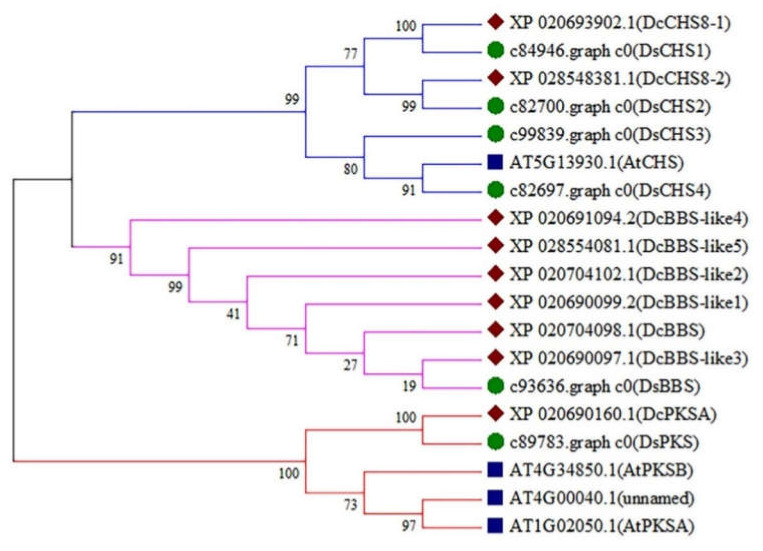
Phylogenetic analysis of type III *PKS* genes. The evolutionary tree was constructed using the neighbor-joining method by MEGA 7.0.

**Figure 4 ijms-23-06780-f004:**
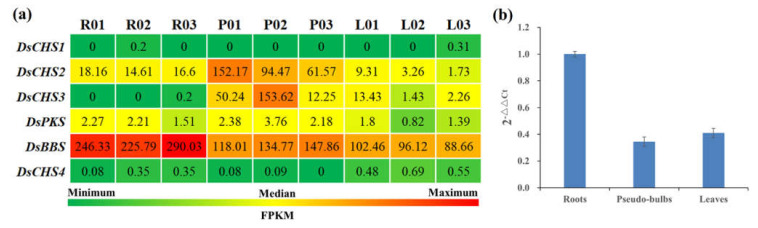
Expression analysis of *PKS* genes in *D. sinense*. (**a**) The heatmap of six *PKS* expressions in different tissues. The heatmap was mapped by Excel using FPKM values. R, roots; P, pseudobulbs; L, leaves. (**b**) RT-qPCR analysis of *DsBBS* gene. The expression levels were calculated by 2^−ΔΔCt^. Error bars indicate standard deviation.

**Figure 5 ijms-23-06780-f005:**
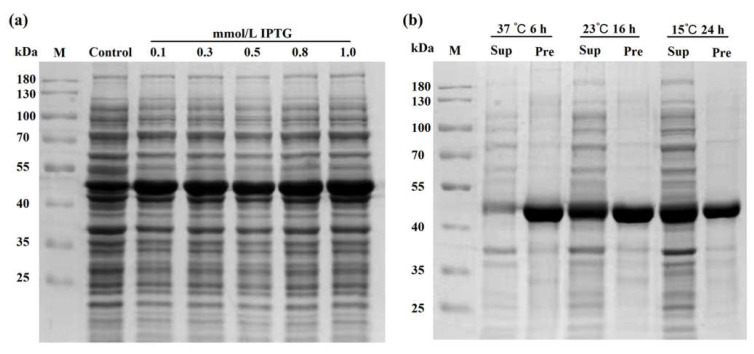
Optimizing prokaryotic expression conditions. (**a**) The optimum inducement concentration of IPTG. (**b**) The optimum inducement time and temperature. Abbreviations: M, marker; IPTG, isopropyl-β-d-thiogalactopyranoside; Sup, supernatant; Pre, precipitate.

**Figure 6 ijms-23-06780-f006:**
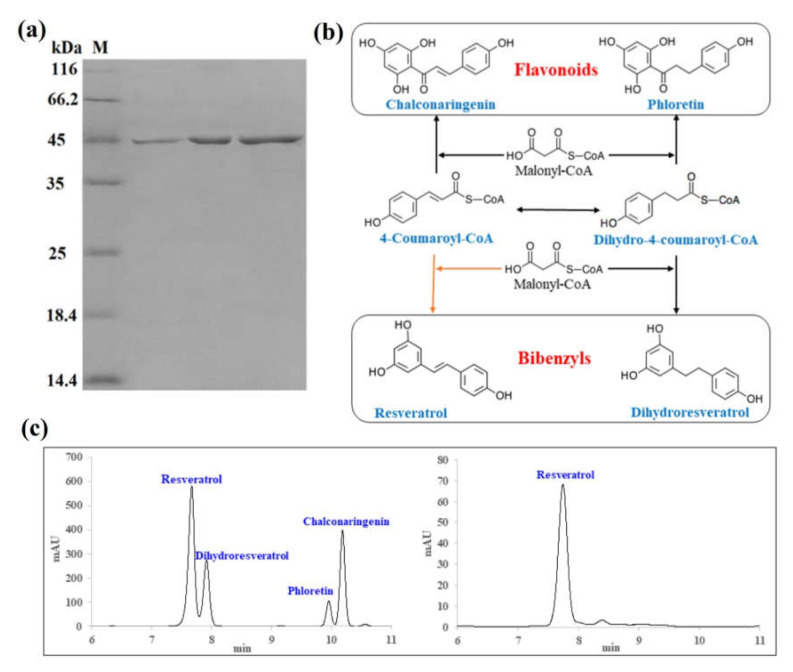
Analysis of the enzymatic reaction products of DsBBS protein. (**a**) Electrophoretic diagram of purified DsBBS protein. (**b**) The beginning steps of flavonoid and bibenzyl biosynthesis. (**c**) HPLC chromatograms of reaction products.

**Table 1 ijms-23-06780-t001:** The type III *PKS* genes in *D. sinense*.

Genes	ID	ORF		
		Start	Stop	Length (nt|aa)
*DsCHS1*	c82697.graph_c0	75	> 764	-
*DsCHS2*	c82700.graph_c0	87	1274	1188|395
*DsCHS3*	c84946.graph_c0	55	1227	1173|390
*DsPKS*	c89783.graph_c0	558	1571	1014|337
*DsBBS*	c93636.graph_c0	101	1273	1173|390
*DsCHS4*	c99839.graph_c0	41	> 838	-

- means an incomplete ORF.

## Data Availability

The transcriptomic data of *D. sinense* have been committed to the NCBI database, such as SRR15112264, SRR15112265, and SRR15112266 for leaves, SRR15112267, SRR15112268, and SRR15112269 for pseudobulbs, and SRR15112270, SRR15112271, and SRR15112272 for roots.

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
