# Peer review of "Characterization of the Key Bibenzyl Synthase in Dendrobium sinense"

_ijms, 2022, doi:10.3390/ijms23126780_

Round 1

Reviewer 1 Report

The article describes identification of the key enzyme in production of resveratrol in Dendrobium sinense, a medicinal orchid endemic in Hainan, China. It was found that the roots of the plant have the highest content of bibenzyls due to the highest expression of the particular (one of six) polyketide synthase gene. The recombinant protein was constructed (45 kDa) and expressed in E. coli culture, producing resveratrol in vitro from 4-coumaryl-CoA and malonyl-CoA as substrates. The study advances the understanding of important phytochemical synthesis in medicinal plants.

The article is generally well written, with adequate illustrations and references. However, it requires minor corrections of grammar, style and spelling. The HPLC analysis (Fig. 6c, subchapter 4.9.) does not mention the mode of detection and identification of the peaks on the chromatogram.

Below are some of the necessary corrections:

Line 13. It should be “roots had higher bibenzyl content than…”. Same on line 358.

Line 29. It should be “orchid”, not capitalized. Same on line 370,

Line 32. It should be “were”, not “are”.

Line 34. Remove “Recently”.

Line 46. It should be Bibenzyls are aromatic compounds with common parent nucleus…”.

Line52. Remove “wide”.

Line 58. What is the meaning of “is followed by…”?

Line 59. The chemical names should not be capitalized here and in many other places (line193, 194. 201, 249, 250)

Line 73-74. Remove the formation of”. It should be “To date,”.

Line 75 Remove “the”.

Line 76. It should be “In this study…”, not “Here”.

Line 77-78 It should be “sequencing” and “performed”.

Line 80. Remove “some”.

Line 85. It should be “revealed”, not “would reveal”.

Line 192. Remove “were”.

Line 193. Remove “As is well known,”.

Line 210. Instead of “the same features as” it should be “similarly to”.

Line 216-218. It should be “analysis of D. sinense has shown that the whole plant is rich in bibenzyls,”. The sentence is unfinished (“was not established”?).

Line 220. Remove “Therefore,”

Line 231. Remove “the”.

Line 238. Remove “Obviously,”

Line 246. It should be “the band”.

Line 279. It should be “was”, not “were”.

Line 281-282. It should be “ “was performed as in [26]”, and “as in [27]”.

Line 301. It should be “was”, not “were”.

Line 311. Cryptic sentence – what was the role of Roche, why in brackets?

Line 331. It should be “E. coli bacteria were cultured…”.

Line 335. It should be “buffer”. Do these bacteria form “thallus”? What is “cracked liquid” in line 337?

Line 350. It should be “HC-C18 column”.

Line 367. It should be “Although”, not “Despite”.

In the References, many Latin names are not italicized.

In summary, this manuscript may be published after addressing the above queries and corrections.

Author Response

Dear reviewers:

Thank you for the comments concerning our manuscript. The main corrections in the paper and the responds to the reviewer’s comments are as following:

  1. However, it requires minor corrections of grammar, style and spelling. The HPLC analysis (Fig. 6c, subchapter 4.9.) does not mention the mode of detection and identification of the peaks on the chromatogram.

Response: Thank you for your comments. The full text is checked and modified. Additionally, comparison of retention time with the standard was used for bibenzyl and chalcone determination (line 366-369).

  1. Line 13. It should be “roots had higher bibenzyl content than…”. Same on line 358. Line 29. It should be “orchid”, not capitalized. Same on line 370.

Response: According to reviewer’s suggestion, these errors were revised (line 13, line 363, line 70, and line 367).

  1. Line 32. It should be “were”, not “are”.

Response: Thank you for your comments. This error was modified (line 35).

  1. Line 34. Remove “Recently”.

Response: “Recently” has been removed (line 37).

  1. Line 46. It should be Bibenzyls are aromatic compounds with common parent nucleus…”.

Response: Thank you for your comments. The statement has been optimized (line 47).

  1. Line52. Remove “wide”.

Response: “wide” has been removed (line 54).

  1. Line 58. What is the meaning of “is followed by…”?

Response: The sentence has been revised, “Subsequently, the CoA esters of these acids are catalyzed to product 4-coumaroyl-CoA and dihydro-4-coumaroyl-CoA” (line 60-61).

  1. Line 59. The chemical names should not be capitalized here and in many other places (line193, 194. 201, 249, 250)

Response: The initials have been changed to lower case (line 61, 193, 194. 252, 254, 256).

  1. Line 73-74. Remove the formation of”. It should be “To date,”.

Response: Modified as required (line 73-74).

  1. Line 75 Remove “the”.

Response: Modified as required (line 75).

  1. Line 76. It should be “In this study…”, not “Here”.

Response: Modified as required (line 76).

  1. Line 77-78 It should be “sequencing” and “performed”.

Response: Modified as required (line 77-78).

  1. Line 80. Remove “some”.

Response: Modified as required (line 80).

  1. Line 85. It should be “revealed”, not “would reveal”.

Response: Modified as required (line 85).

  1. Line 192. Remove “were”.

Response: Modified as required (line 192).

  1. Line 193. Remove “As is well known,”.

Response: Modified as required (line 193).

  1. Line 210. Instead of “the same features as” it should be “similarly to”.

Response: Modified as required (line 210).

  1. Line 216-218. It should be “analysis of D. sinense has shown that the whole plant is rich in bibenzyls,”. The sentence is unfinished (“was not established”?).

Response: We are very sorry for this wrong expression. The statement has been modified (line 216-217).

  1. Line 220. Remove “Therefore,”

Response: Modified as required (line 221).

  1. Line 231. Remove “the”.

Response: Modified as required (line 231).

  1. Line 238. Remove “Obviously,”

Response: Modified as required (line 239).

  1. Line 246. It should be “the band”.

Response: Modified as required (line 247).

  1. Line 279. It should be “was”, not “were”.

Response: Modified as required (line 281).

  1. Line 281-282. It should be “ “was performed as in [26]”, and “as in [27]”.

Response: Modified as required (line 283-284).

  1. Line 301. It should be “was”, not “were”.

Response: Modified as required (line 303).

  1. Line 311. Cryptic sentence – what was the role of Roche, why in brackets?

Response: We are very sorry for this wrong expression. The statement has been modified (line 312-313).

  1. Line 331. It should be “E. coli bacteria were cultured…”.

Response: Modified as required (line 334).

  1. Line 335. It should be “buffer”. Do these bacteria form “thallus”? What is “cracked liquid” in line 337?

Response: Modified as required (line 340).

  1. Line 350. It should be “HC-C18 column”.

Response: Modified as required (line 353).

  1. Line 367. It should be “Although”, not “Despite”.

Response: Modified as required (line 372).

  1. In the References, many Latin names are not italicized.

Response: Modified as required.

We tried our best to improve the manuscript and made some changes in the manuscript. These changes will not influence the content and framework of the paper.

We appreciate for Editors/Reviewers’ warm work earnestly, and hope that the correction will meet with approval.

Once again, thank you very much for your comments and suggestions.

Yours sincerely,

Niu Jun

Reviewer 2 Report

Dear authors, 

Please format the latin name of the species in title accordingly!

Line 34-35- revise the sentence, dont use "hot topic"

line 43 - mining? or do you mean extraction?

Figure 1 - are the concentrations really that high? its seems like these alues are overexagerated?

The results are well written and informative, good representation of findings.

materials and methods are well written and the experiments could be reproduced.

references are well formatted and chosen well suiting the topic of the article, however, some of the references are a bit too old.

Author Response

Dear reviewers:

Thank you for the comments concerning our manuscript. The main corrections in the paper and the responds to the reviewer’s comments are as following:

  1. Please format the latin name of the species in title accordingly!

Response: the latin name was written in italics.

  1. Line 34-35- revise the sentence, dont use "hot topic"

Response: According to reviewer’s suggestion, the sentence was changed to “The studies of Dendrobium species tend to focus on exploration and pharmacology of bibenzyls.”

  1. line 43 - mining? or do you mean extraction?

Response: Modified as required.

  1. Figure 1 - are the concentrations really that high? its seems like these alues are overexagerated?

Response: The content accounted for 1.31%, 0.62%, and 0.72% of the dry weight of roots, pseudobulbs, and leaves, respectively. This result was similar to the previous report of Dendrobium denneanum, and 0.97% of total bibenzyl was detected in the whole plant (Fang et al, 2014). Thus, the detection results of this study should be relatively reliable.

Reference: Fang, J.; Houlin, X.; Ping, H.; Xiaoyan, H.; Zijun, N.; Jianjun, T.; Tingmo, Z., Determination of Bibenzyl Compounds Content in Dendrobinm denneanum by UV-Visible Spectrophotometry. J. Chengdu Univ. TCM 2014, 37, (01), 7-10.

  1. references are well formatted and chosen well suiting the topic of the article, however, some of the references are a bit too old.

Response: Thank you for your comments. We have updated some of the references.

We tried our best to improve the manuscript and made some changes in the manuscript. These changes will not influence the content and framework of the paper.

We appreciate for Editors/Reviewers’ warm work earnestly, and hope that the correction will meet with approval.

Once again, thank you very much for your comments and suggestions.

Yours sincerely,

Niu Jun